# Exploring Benefits and Barriers of Plant-Based Diets: Health, Environmental Impact, Food Accessibility and Acceptability

**DOI:** 10.3390/nu15224723

**Published:** 2023-11-08

**Authors:** Giulia Viroli, Aliki Kalmpourtzidou, Hellas Cena

**Affiliations:** 1Laboratory of Dietetics and Clinical Nutrition, Department of Public Health, Experimental and Forensic Medicine, University of Pavia, 27100 Pavia, Italy; aliki.kalmpourtzidou01@universitadipavia.it (A.K.); hellas.cena@unipv.it (H.C.); 2Clinical Nutrition Unit, General Medicine, ICS Maugeri IRCCS, 27100 Pavia, Italy

**Keywords:** plant-based diets, sustainable diets, food affordability, food acceptability

## Abstract

Unhealthy dietary patterns are directly linked to the current Global Syndemic consisting of non-communicable diseases, undernutrition and climate change. The dietary shift towards healthier and more sustainable plant-based diets is essential. However, plant-based diets have wide intra differences; varying from vegan diets that totally exclude meat and animal products to traditional ones such as the Mediterranean diet and the new Nordic diet. It is acknowledged that plant-based diets may contribute simultaneously to improving population health as well as to decreasing the environmental impact of food systems. Evidence from cohort and randomized-controlled trials suggests that plant-based dietary patterns have beneficial effects on bodyweight control, cardiovascular health and diabetes prevention and treatment. On the other hand, micronutrient requirements may not be met, if some plant-based diets are not well-planned. Additionally, studies showed that lower consumption of meat and animal products results in lower environmental impacts. Consequently, plant-based diets could be a key factor to increase diet sustainability. This narrative review addresses the advantages of adherence to plant-based diets on human and planetary health considering strains and barriers to achieve this dietary transition, including cultural acceptability and affordability factors. Finally, potential intervention and policy recommendations are proposed, focusing on the update of current national food-based dietary guidelines.

## 1. Introduction

In the 21st century, the world faces climate change, obesity and undernutrition identified as the Global Syndemic introduced by the Lancet Commission. The Global Syndemic is defined as a cluster of pandemics representing a confluence of socioeconomic forces leading to adverse effects on human health and environment [1].

Although some progress has been made counteracting undernutrition, hunger re-mains a critical issue for 150 million children affected by stunting in 2019 [2]. Moreover, the prevalence of overweight and obesity is high, affecting 2 billion people globally, especially in low and low-middle income countries. Low and low-middle income countries are also experiencing the “nutrition transition”, which is described as a shift from traditional diets to “Western” diets characterized by high saturated fats, sodium, sugar and meat intake [2,3,4]. The adoption of a Western-style diet leads to unbalanced nutrient intake, resulting in measurable adverse health effects including obesity, non-communicable diseases (NCDs) and early all-cause mortality [4]. Globally, unbalanced dietary patterns are ranked as top risk factors for mortality and disability-adjusted life-years lost, with 11 million deaths and 255 million disability-adjusted life-years lost, mainly because of cardiovascular diseases (CVDs), type 2 diabetes and cancer [5].

On the other hand, climate change poses a threat to human health by causing biodiversity loss, species extinction and degradation of natural resources [6,7]. Temperature changes are expected to reduce agricultural productivity, micronutrient food content, and food availability, thus increasing food prices and enforcing adoption of unhealthy dietary patterns [7,8,9]. 

Diet composition significantly influences the environment through the mediation of food systems [10], one of the largest contributors to climate change, the impact of which is likely going to increase due to population growth [11,12,13]. Unsustainable agricultural expansion, driven in part by the non-adherence to balanced diets (diets based on whole grains, legumes, fruits and vegetables, nuts and seeds, and animal-source foods produced in resilient and sustainable systems), increases ecosystem and human vulnerability and leads to competition for land and/or water resources [7].

Consequently, nutrient-balanced diets based on plant-based food products could be a key link between human health and environmental sustainability. For example, in 2019 the EAT-Lancet Commission Panel defined the Planetary diet composed mainly of vegetables, fruit, whole grains, legumes, nuts and unsaturated fats, moderate or small amounts of fish and poultry, and no or low consumption of red meat, highly processed foods, processed meat, added sugars, refined cereals and starchy vegetables [14]. This diet could potentially meet the global environmental, nutritional and health targets [14]. 

Dietary patterns that emphasize the consumption of plant-based foods while eliminating most or all animal products are defined as plant-based diets (PBDs) [13,15,16]. There are different types of PBDs: pesco-vegetarian diets that eliminate animal products except for fish and seafood; lacto-ovo-vegetarians that exclude meat and fish but not dairy products and eggs (lacto-vegetarian and ovo-vegetarian variants eliminate eggs and dairy products, respectively); vegan diets that exclude meat and all animal food products and derivatives including honey. Plant-based diets such as the Mediterranean [17], Dietary Approaches to Stop Hypertension (DASH) [18], the New Nordic [19] and the Planetary diet [14] may allow the consumption of small quantities of meat and animal food products. The Mediterranean diet [17] is a dietary pattern that incorporates traditional living habits of people from countries surrounding the Mediterranean Sea. It is based on fruits, vegetables, wholegrains, legumes, a moderate consumption of dairy and fish and a low consumption of meat and sweets. Extra virgin olive oil as the main fat source and local herbs for seasoning are recommended [17]. The DASH diet was developed in the USA to prevent and manage hypertension. It is a dietary pattern based on vegetables, fruits, and wholegrains and includes fat-free or low-fat dairy products, fish, poultry, beans and nuts. The amount of sodium is highly recommended to be limited [18]. The New Nordic Diet is rich in fruits, berries, vegetables, legumes, low-fat dairy products and fatty fishes that are readily obtained in Nordic countries [19].

Furthermore, dietary behaviors are shaped by different drivers, from personal values, beliefs and preferences to family and friends, school, workplace, communities, urban design, food availability and transportation. These drivers are in turn shaped by policies, economic incentives and social norms [1,20,21]. Recognizing the role of social, economic and environmental factors, as well as genetic ones, is the first step for the implementation of policies to support individual health.

This narrative review aims to address benefits and barriers of PBD adoption in accordance with four dimensions: health, environmental impact, social and economic effects. These variables are considered by the Food and Agricultural Organization in the definition of a sustainable diet, as a “nutritionally adequate, healthy, safe, culturally acceptable, economically fair, accessible and affordable, and protective and respectful of biodiversity and ecosystems” diet [22]. Therefore, this paper will explore if a PBD can meet the health, environmental impact, social and economic criteria of a sustainable diet. Finally, to ensure PBD adoption, health policy recommendations are provided. 

## 2. Methods: Literature Search Strategy

The literature search was conducted on the PubMed database. To investigate the health effects of PBDs, the search terms used were: plant-based diet, vegetarian, vegan and health outcomes such as obesity, diabetes, cardiovascular disease and mortality. Studies in the English language, peer-reviewed, available in full text, but also meta-analysis and systematic reviews of randomized controlled trials or longitudinal cohort studies design were included. The environmental impact search terms used related to diets were: environmental impact and plant-based diet or dietary change. Environmental impact of current diets and predicted dietary scenarios were considered adequate for this manuscript as long as they considered multiple domains of environmental impact indicators (e.g., greenhouse gas emissions (GHGEs), land use, water use). In addition, since the aim of the paper is to connect healthiness, environmental impact, food affordability and acceptability of diets, the focus was on papers investigating multiple scenarios. Affordability and acceptability were searched as: plant-based diet and cost, affordability, acceptability. Health policies and other references were selected following a snowballing approach.

## 3. Health Benefits of Plant-Based Diets

Adherence and interest to plant-based dietary patterns are increasing [23]. Therefore, the assessment of the relationship between PBDs and health is important to define adequate, safe and valid nutritional indications for individuals and the general population. 

### 3.1. Effect of Plant-Based Diets on Body Weight Control and Obesity

A high intake of red and processed meat has been linked to increased weight, cancer and type 2 diabetes incidence due to their content in saturated fats and salt [24,25,26,27], while a regular fish and seafood consumption is recommended thanks to their high content of ω-3 polyunsaturated fatty acids that reduce the risk of cardiac death [28]. Indeed, national and international guidelines recommend at least 250 mg/day or at least two servings/week of oily fish [28]. 

Plant-based diets may be useful for the prevention of weight gain and control of body weight [29,30]. For instance, vegetarians showed a lower body weight compared to non-vegetarians in cross-sectional studies [31,32]. A systematic review of prospective cohort studies [33] which included nine cohorts from different countries (USA, Finland, UK, Bulgaria, New Zealand, Australia, Spain, the Netherlands, South Korea and Belgium) found that a high adherence to PBDs was associated with lower body adiposity. A meta-analysis of 12 randomized-controlled trials on participants with overweight or obesity [34] analyzed weight reduction achieved by following meat-free dietary patterns (vegan or lacto-ovo-vegetarian diets) compared to other dietary interventions (e.g., the American Diabetes Association-recommended diet, the diet supported by the National Cholesterol Education Program, and the Atkins diet). The authors found that participants assigned to the meat-free dietary patterns lost significantly more weight than those assigned to the omnivorous diet groups (−2.02 kg; 95% CI: −2.80 to −1.23). Noteworthily, the results were attenuated over time (one year of follow-up). Several studies [35,36,37,38] assessed the effectiveness of PBDs on weight loss but mostly reported short-time results and, therefore, with limitations. Since obesity is a chronic and multifactorial disease, the necessary changes to dietary habits should be sustained for long periods to prevent rebound after weight loss.

### 3.2. Plant-Based Diet Effect on Blood Lipids and CVD Risk

Prevention and treatment of obesity are important since obesity is a driver towards NCDs including CVDs and type 2 diabetes. The meta-analysis by Koch et al. (2023) [39] evaluated 30 randomized controlled trials of vegetarian and vegan diets compared to omnivorous diets (consuming all food groups). The authors found a significant improvement for total cholesterol (−0.34 mmol/L, 95% CI: −0.44, −0.23), low-density lipoprotein (−0.30 mmol/L, 95% CI: −0.40, −0.19) and apo-B levels (−12.92 mg/dL, 95% CI: −22.63, −3.20) while there was no improvement for triglycerides (0.06 mmol/L; 95% CI: −0.01, 0.13). Benefits on low-density lipoprotein, total cholesterol and high-density lipoprotein were reported in other meta-analyses of prospective cohort studies [40,41,42], and similar results were observed for triglycerides [39,40]. A meta-analysis of 13 unique prospective cohorts (including Adventists, Nurses’ Health, Oxford, Tzu Chi cohorts), with a follow-up ranging from 5 to 28 years and a sample size ranging from 1724 to 422,791 participants across all outcomes, assessed the CVD risk in vegetarians and vegans compared to controls [43]. The authors observed that vegans and vegetarians had a lower CVD risk. In particular, vegetarians had the highest reduction (15%) in the relative risk of CVDs compared to non-vegetarians. However, contrasting results were observed in similar meta-analyses of prospective studies [44,45,46]. In addition, a recent meta-analysis of prospective cohorts with a follow-up period ranging from 3.5 to 32 years including 715,128 participants confirmed that a high intake of plant-based proteins compared to a low intake was significantly associated with a lower risk of CVD-related mortality (*p* = 0.001) and all-cause mortality (*p* = 0.003) [47].

### 3.3. Plant-Based Diet Effect on Blood Glucose and Type 2 Diabetes Risk

Similarly, cumulative evidence suggests that vegetarian diets may have beneficial effects compared to omnivorous diets on the risk for type 2 diabetes, which seemed independent from body mass index reduction [48,49,50]. A meta-analysis of randomized controlled trials found that in eight studies, vegetarian diets compared to omnivorous diets and National Diabetes Guidelines were associated with a greater reduction in HbA1c levels [51]. Papamichou et al. (2019), reported reduction in HbA1c levels from adherence to vegetarian and Mediterranean dietary patterns on type 2 diabetes prevention and treatment [52]. Wang et al. (2023) [53] found higher improvements in HbA1c levels from vegetarian diets compared to actual and conventional energy-restricted diets related to diabetes [53]. Moreover, a systematic review by Toumpanakis et al. (2018) [51] on adults with type 2 diabetes found PBDs significantly improved HbA1c levels when compared to dietary guidelines for diabetes. A long randomized controlled trial (74 weeks) on individuals with type 2 diabetes found a reduction in HbA1c levels from American Diabetes Association guidelines and a low-fat vegan diet [54].

Despite the potential health benefits of PBDs, the control diet is often not described in trials, but named as “non-vegetarian”, “usual diet” or “omnivorous diet”. Thus, the evaluation of the health impact of different PBDs needs a detailed definition of control diets, too.

### 3.4. Nutrient Adequacy

Regarding nutrient adequacy, subjects following a vegetarian or vegan dietary patterns are at risk of vitamin B12, D, calcium, iron, niacin (B3), iodine, selenium and zinc deficiencies [55,56,57,58,59]. However, well-planned vegetarian diets that include a wide variety of plant-based food products may provide adequate micronutrient intake [55]. Nevertheless, people experiencing diet transition, especially in low-income countries, may encounter some barriers to shift towards healthy PBDs, due to lack of education and food insecurity [60]. Since evidence has shown that PBD adoption can improve human and planetary health, barriers need to be identified and taken into account for future health policies. For instance, modeling studies that calculated an optimal diet considering both healthiness and environmental impact constraints confirmed that adherence to PBDs could lead to vitamin B12, zinc and calcium deficiency [61,62,63]. Concerns about nutrient adequacy were reported also for the Planetary diet [64]. Thus, to account for an adequate nutrient intake, future dietary supplementation and food fortification-targeted health policies are required.

Moreover, nutrient adequacy of diets depends on specific combinations of food groups. Considering the variety of products on the expanding plant-based alternatives market [65], there are plant-derived foods such as juices, sweetened drinks, refined grains, sweets, French fries, and pre-fried plant-based meat substitutes which are associated with a higher risk of CVDs [66]. A dietary pattern that includes the consumption of these types of food items on a daily basis is defined as an unhealthy PBD [67,68,69]. Unhealthy PBDs have indeed been correlated to a higher risk of dyslipidemia [67], incidence of type 2 diabetes [68] and coronary heart disease [69].

## 4. Environmental Impact of Plant-Based Diets

Western-style diets, based on animal products, result substantially in high GHGEs, freshwater and energy use [70]. Several studies calculated the reduction in environmental impact indicators such as GHGEs, land use and freshwater use [11,71,72,73,74,75] obtained from the shift towards PBDs, with GHGEs being studied the most. Overall, the results suggest that the higher avoidance of animal food products results in lower environmental impacts. Aleksandrowicz et al. (2016) [10] reviewed the environmental impact of different PBDs characterized by different quantities of animal products, such as pescatarian diets, the Mediterranean diet, healthy dietary guidelines, vegetarian and vegan diets. The authors found that GHGEs, land and water use were reduced by 22%, 28% and 18%, respectively, across all PBD types. Vegan scenarios had the greatest median reductions for GHGEs (−51%) and land use (−45%) while the lowest performances were noticed for dietary scenarios with balanced energy intake or partial meat replacement with dairy products. Vegetarian scenarios instead showed the largest benefit (−37%) for water use, while an increase resulted for vegan scenarios, although the number of studies was limited and results in other studies assessing the vegan diet impact on water use were controversial [74,75,76].

To meet both environmental impact and health benefits, a growing number of studies use optimization modeling to define optimal diets that do not exceed environmental boundaries and simultaneously reduce the risk of diet-related NCDs, ensuring an adequate nutrient intake [61,62,77,78]. Overall, evidence suggests that shifting to a healthy diet is not only good for human health but also for the environment [13]. Chaudhary et al. (2018) [62], analyzed nutrition–environmental outcomes of dietary changes of 156 countries towards three different scenarios: global guidelines on healthy eating and energy intake, vegan and vegetarian diets. Results indicated that dietary changes toward lower animal and more plant-based foods consumption improved nutrient intake and reduced GHGEs and freshwater use in North America and Europe. Regarding high-income countries, to reach an adequate nutrient intake in other countries (such as in South Asia), an increase in the GHGEs and freshwater use was necessary. Thus, dietary change scenarios did not always lead to both nutritional and environmental benefits for all countries.

To define clearly sustainable diets, environmental impact, healthiness and nutritional adequacy need to be considered in conjunction with the food affordability and the cultural acceptability of diets [22,62,79].

## 5. Affordability of Plant-Based Diets

Food choices are influenced by individual, cultural, social and environmental factors, but affordability is also a major factor [80]. Unhealthy dietary patterns are more prevalent in low socio-economic status subgroups compared to wealthier subgroups [81,82,83,84]. Nevertheless, food price may not be the only barrier; education, taste preferences and cultural eating habits [85,86] play a role, too.

The cost of the diet transition can be calculated in different ways. Studies often compare different prices of the same amount of food items by defining them as “healthier” or “less healthy” (e.g., more vs. less healthy grains), while other studies examine the price differences of “healthier” compared to “less healthy” overall dietary patterns. In addition, food price can be estimated as the price of a unit portion per day (cost/average portion) or per calories (cost/kcal). To add complexity, the relative affordability of different foods also depends on the relation between local food prices and household food budgets [87,88].

Despite the different methodologies, the cost of the transition from current diets to a PBD would be more affordable in high-income countries, while in low-middle income and low-income countries it would be more expensive [61,62,89,90,91]. Kalle Hirvonen and colleagues (2020) [92] calculated the Planetary diet affordability, reporting that it would be unaffordable for almost 1.58 billion people living in low-income countries, and the cost of the diet would disproportionately affect mean daily household income per capita (from 6% to 89% in low-income countries).

Rao et al. (2013) [88] found that the consumption of a “healthier” versus “less healthy” diet (including Mediterranean-type diets rich in fruits, vegetables, fish and nuts versus diets rich in processed foods, meats and refined grains) had a higher price difference per person than the actual dietary pattern. The cheaper prices of a “less healthy” diet could be partially explained by the low-cost of empty-calories foods [1]. To ensure a healthy, environmentally sustainable and affordable diet for all, the production of planet-friendly food items that are both affordable and nutrient-rich represents a challenge to agricultural food systems. 

## 6. Acceptability of Plant-Based Diets

The acceptability of foods depends on environmental factors as well as convenience, availability, social interactions, sensory attributes, culture and context [93]. Plant-based diets are generally considered acceptable in low- and low-middle income countries because of cultural, financial, ethical and/or religious reasons, while in high-income countries people often consider meat an important part of a meal and the attitude towards vegetarian/vegan diets are to be inconvenient, difficult to prepare and less enjoyable than omnivorous diets [94]. Nonetheless, trends are changing and currently the number of vegan/vegetarian people in high-income countries is growing [23], while a nutritional transition in low- and low-middle income countries, such as China and India, is taking place [4] with an increasing consumption of meat and animal products due to the per capita income growth [2,3].

Optimization studies showed that a trade-off between environment, nutrient adequacy and cost can be achieved, with some criticality about the diet affordability as mentioned before [95,96]. However, the optimized diets in some cases differed greatly from the current dietary habits of the populations; hence, optimized diets may not be accepted and followed. For example, the Planetary diet requires a drastic reduction in consumption of meat, eggs, tubers, and refined grains, while vegetable oils, legumes, and whole grains would be significantly higher compared to mean consumption worldwide [14]. 

As underlined by Perignon [91], the diet optimization approach could identify the optimal diets which fulfil lower environmental impact, better nutritional quality, and no additional costs, ensuring at the same time an acceptable deviation from the current diet [97,98,99,100]. Vieux et al. (2020) [99] applied this method using data from dietary surveys of five European countries (France, Finland, Italy, Sweden, and United Kingdom). The authors found that one individual out of five already followed a plant-based sustainable diet. However, the optimized diet compared to the observed average European diet included a much higher amount of plant-based products, with a substantial difference of fruit and vegetable consumption and a lower content of sweets and alcoholic drinks. On the other hand, the total quantity of animal-derived products was the same for both diets, but the type of animal products was distributed slightly differently. In particular, the optimized diet compared to the average diet of the five European countries included less ruminant meat, processed meat and offal, a smaller quantity of composite dishes containing animal products but more dairy products and slightly more fish, while the quantities of eggs and poultry were similar. Additionally, to achieve environmentally sustainable, healthy and affordable diets, it is important that energy intake does not exceed the individual total energy requirements [91]. Consequently, these results showed that it is possible to reduce dietary GHGEs while increasing the nutritional quality without eliminating whole food categories, such as meat and animal products, offering the opportunity even to those who are not vegans to follow a PBD and still consider it acceptable.

Long-term randomized controlled trials that investigated the health impact of PBDs compared to a balanced control diet showed that the acceptability of the plant-based dietary pattern was not different from the control diet [51,54,95,101], even though in one study the vegan group rated their diet as more difficult to prepare than their usual diets [101]. At a population level, those results could highlight that if people improve their food knowledge, as well their skills to cook and organize tasty meals, the adherence to PBDs could be adopted by more people.

The promotion of a sustainable plant-based dietary pattern requires context-specific solutions because of the diversity of dietary habits, food cultures and environmental matters. Familiarity is one of those factors that drive consumer acceptance [102] and, hence, the emphasis on local, known foods may lead to a higher acceptability and, at the same time, to environmental benefits [103]. The Mediterranean diet model, for instance, is not just a plant-based dietary pattern associated with health outcomes [17]; it is also aligned to the cultural richness linked to the territory, the conviviality, and the society that is transformed into a real act of relationships and sharing though food [104]. The Mediterranean diet exerts health benefits also in non-Mediterranean populations [105,106,107], incorporating foods that are culturally acceptable, locally produced and accessible, but with a similar nutritional profile to those prominent in a Mediterranean diet [108,109]. As for the study on the Planetary diet [14], the authors underlined that plant-based dietary patterns need to be adopted considering the culture and personal preferences of the local populations and food item availability and cost.

## 7. Food-Based Dietary Guidelines and Recommendations for Policy Makers

In 1992, the ‘World Declaration and Plan of Action for Nutrition’ convened by the Food and Agriculture Organization and World Health Organization, recommended government dissemination of a dietary guidance to their countries [110,111] to provide context-specific advice and principles on healthy diets and lifestyle. The dietary guidance should respond to public health and nutrition priorities, food consumption patterns, sociocultural influences, food composition data and food accessibility. Therefore, alignment of the Food-Based Dietary Guidelines (FBDGs) beyond health-related issues to the wider environmental and social framework would allow the development of coherent policy that could contribute not just to personal, community and health issues, but also to global health [1,112,113].

Springmann et al. (2020) quantitatively analyzed the environmental and health impact of 85 FBDGs to assess if they were compatible with planetary environmental boundaries and mortality from chronic diseases [114]. The authors found that, considering the Paris Agreement limits, the GHGEs of FBDGs were exceeded by 140% on average. Only 13% of FDBGs were compatible with a food-related GHGEs pathway of limiting global warming below 2 °C. Furthermore, 22% of them were in line with global land-use targets and 33% with freshwater-use targets, while 66% of the FBDGs decreased the risk of NCDs and premature mortality. In addition, the habitual diet of the population was on average far from the national FBDGs. Consequently, even if populations improved their diets to fully align with FBDGs, the reduction in GHGEs would be on average 13%.

Currently, the inclusion of the sustainability constraints in FBDGs is in progress in several countries. According to James-Martin et al. (2022) [115], the FBDGs of 37 countries mention environmental sustainability, although they represent approximately 17% of the world population. Sociocultural-associated accessibility of food guiding principles was reported by 38% of FDBGs in the background session, and 6% reported sociocultural framework in consumer documents. Belgium [116] and Sweden [117] were the countries which mentioned the highest number of guiding principles for sustainable healthy diets in consumer and background documents.

For these reasons, the update of FBDGs is required for ensuring their alignment with environmental constrains. Future FBDGs should also take into account the context and the socio-cultural framework, as well the accessibility, of food groups [118]. Additionally, the number of FBDGs using food processing-related terms (such as ‘processed foods’ and ‘processed meats’) has been growing over the past 20 years but is still a small number compared to the total of FBDGs [119]. Dietary guidance about food processing should be implemented in FBDGs to provide a foundation for consumer understanding. Changes in lifestyle, such as missing cooking skills and change of food-purchasing behaviors like the consumption of ready-to-eat meals, make it necessary to deal with these foods and behavioral changes [119,120,121].

To facilitate and improve the adherence to FBDGs and to have a real effect on population food consumption, FBDGs need to have clear links to food policies that are implemented in school and hospital canteens, public procurement, advertising regulations and industry standards [122,123]. In addition, efficient actions from a multidisciplinary technical team, which includes social, economic, human food chain professionals and health professionals, is essential [124,125].

Besides the information included in the FDBGs, the communication and dissemination of results is equally important [126,127,128,129]. Due to the rapidly growing and evolving nutritional literature, identifying the most potentially impactful method to improve nutritional knowledge and communication impact is challenging [126,127]. Incorrect and conflicting nutritional and health messages communicated to a broad audience lead often to misunderstanding of the information by consumers [126], and therefore, the dissemination of the FBDGs to the most common media communication should be recommended. The principles of FBDGs should be expanded also to all health professionals who advise on nutrition. Indeed, incomplete knowledge was recognized as one of the main barriers to PBD adoption [102,130]. 

Awareness towards PBDs should go along with simple, practical information about the preparation of tasty plant-based meals [131,132], and educational programs should be set especially for children in primary and secondary schools [133]. An effective and expanding way to educate people to eat more vegetables and fruit is through the construction of urban gardens and farms [134,135]. Involvement in activities such as gardening, harvesting, cooking, preparing dishes and tasting the products has shown to increase vegetable consumption and diet diversity of children attending a summer camp in the USA [135]. Similarly, the exposure to different types and textures of foods could reduce neophobia, which is another barrier to the consumption of plant-based products and diet diversity, especially in children [102]. Also, environmental interventions in urban settings, such as the use of community gardens, showed to improve the diet quality of adults and to increase food security [136,137,138,139]. The construction of rooftop gardens in the center of an Italian city was estimated to be able to cover 64 % of the city’s fresh vegetable requirements [138].

Even if sustainability awareness is a crucial element, it is not sufficient to lead towards a behavioral change. Behavioral changes require a combination of regulatory, financial, technological and environmental-related actions [139]. Taxation of unhealthy foods [140] and subsidies for healthier foods [141] for low-income individuals could be an evidence-based intervention to balance price differences [142] and improve the adherence to a more plant-based sustainable diet. Indeed, nutrition knowledge without food accessibility is a key barrier to healthy food consumption [84]. For instance, sugar-sweetened beverage taxes are associated with lower sales of the taxed products, and it has been implemented in more than 45 countries, with Mexico being the first country to adopt this policy [143,144].

Hence, actions aimed at changing people’s behavior need to co-evolve with the changes of food systems, combining improvements in technologies, management and reductions in food loss and waste [145]. Moreover, the food industry should encourage the development of new plant-based products with healthy, nutritionally adequate, tasty and affordable profiles in line with the country culture and local production [142]. Food innovation actors should also consider fortification for major micronutrients due to the potential deficiency risk for vegetarian and vegan consumers [146].

Research beyond PBD impacts on GHGEs, water and land use should assess the environmental impact of diets and food systems on other environmental indicators such as eutrophication, land-use change and biodiversity loss [11]. In addition, the development of reliable sustainable diet indexes is important for the valid assessment of diet sustainability [147]. Indeed, the construction of interdisciplinary metrics of diet sustainability could allow tracking the progress of guideline influences, evaluate current dietary pattern sustainability and drive policy responses [147]. Several indexes based on one or more sustainable indicators have already been developed [147,148,149,150,151,152,153]. For example, the Planetary Health Diet Index [154] was created to measure the adherence to the Planetary diet. It is composed of 16 components, each of which could be ascribed with a maximum of 10 or 5 points, resulting in a total score ranging from 0 to 150 points. The final score is associated with dietary quality and lower carbon footprint. One limitation of this index is that only a food-frequency questionnaire method was used to estimate individual’s food intake, which may introduce some memory bias and takes a relatively long time to be completed [155]. The SHED index (Sustainable HEalthy Diet Index) [149] is a 30-item web-based questionnaire developed to assess the nutritional, environmental and sociocultural aspects of individual diets. It does not require filling out a food-frequency questionnaire or other questionnaires of nutritional assessment, but it does not consider the economic and preference domains and was validated on a young population (20–45 years old). Thus, the harmonization of a uniform set of parameters with the scopes that integrate health, environmental, economic and social aspects when defining sustainable diets will be important. Acceptability and accessibility of PBDs should be evaluated in long-term trials, too.

Future inclusive dialogues between scientists, policymakers and other stakeholders including the different actors in food systems are needed to provide solutions to face the planetary and people health challenge. The promotion of a common food policy framework at the various levels of governance is essential, with local governance stakeholders being vital as food-policy actors. 

In conclusion, PBDs could be nutritionally adequate, healthy, culturally acceptable, affordable and respectful towards the environment. However, to ensure this sustainable dietary transition from current dietary patterns to sustainable plant-based dietary patterns, barriers should be taken in account and tackled. Substantial actions and interventions could include the update and effective dissemination of FDBGs, urban gardens and farms establishment and promotion, food fortification, innovation of plant-based alternatives based on local foods and taxation of unhealthy food products.

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
