# Peer review of "Exploring Benefits and Barriers of Plant-Based Diets: Health, Environmental Impact, Food Accessibility and Acceptability"

_nutrients, 2023, doi:10.3390/nu15224723_

Round 1
Reviewer 1 Report
Comments and Suggestions for Authors
The manuscript entitled “Exploring benefits and barriers of sustainable diets: health, environment, food accessibility and acceptability” authored by Viroli et al aims to address in a narrative review the health and environmental benefits of a recommended shift towards a (predominantly) plant-based diet pattern.
While the objective of the paper is of interest as the topic as such is highly relevant, there are major limitations with the manuscript that need to be addressed by the authors.
For instance, in the title the term ‘sustainable diets’ is used, whereas the text mostly refers to plant-based diets. It would help in better understanding the focus of the paper if the terms ‘plant-based diets’ and ‘sustainable diets’ would be defined already in the introduction part. Is ‘sustainable diets’ seen as a synonym of ‘plant-based diets’? if so, this should be explained. Further, it would be more correct if when referring to the Mediterranean diet, the DASH diet and the Nordic diet, the term predominantly plant-based diets would be introduced as these diets typically contain low amounts of animal foods.
The manuscript is an information overload, which is not always described in a clear, straightforward way. Many very-long sentences make it further difficult to follow the content and are exhausting. It would be helpful if the scope of the paper was better outlined. Furthermore, it is not clear what take away messages the authors want to give the reader.
Specific comments to the authors:
- General remark: There are many abbreviations used of which some are not explained at first time use; this makes reading the manuscript in parts difficult. The authors could be stricter with the use of abbreviations and ideally also limit their use.
- It would be worthwhile to explain why the authors have chosen the narrative review format, what justified the choice of this methodology? Also, a short part describing the search strategy and criteria and which type of literature was selected would be beneficial.
- 1. Introduction: As mentioned above, introducing, describing, and defining the later use of the terms ‘plant-based diets’ and ‘sustainable diets’ should be introduced.
- 2. Health benefits of plant-based diets: This part needs to be better structured e.g., in the following way: Describing in the necessary detail the various dietary patterns, and then better clustering the health benefits with sub-headers, e.g. , effect on body weight control and obesity, on blood lipids and CVD risk and on blood glucose and diabetes risk and lastly on intake of critical nutrients such as minerals and vitamins.
- 3. Environmental impact of plant-based diets: This part is less focussed and bloated with information which strictly does not seem to fit well under the headline, e.g., the part on micronutrient intake and the description of an unhealthy-plant-based diet, While the latter is an important aspect, it should be better addressed in more detail under Part 2 Health benefits and potential risks.
- 4. Affordability of plant-based diets and 5. Acceptability of plant-based diets: These parts are difficult to understand as overloaded with facts, a better structure, and clear key messages of what the authors want to address here would make it easier to follow.
- 6. Recommendations for policy makers: This part also needs better structure and should for instance firstly explain what is already achieved, e.g., which food-based dietary guidelines in which countries already address environmental aspects like sustainability of dietary patterns followed by what could be further done. And a clear conclusion paragraph should be provided.
Comments on the Quality of English Language
English language editing is required.
Reviewer 2 Report
Comments and Suggestions for Authors
In this narrative review the authors describe a very relevant topic. The idea of summarizing key findings in the area of plant-based diets is welcomed.
The structure of the narrative makes sense, although the last paragraph seems broader than recommendations for policy makers. The abstract mentions potential intervention and policy recommendations, so I would change the title of that last paragraph to reflect this. In the title I would say environmental impact to avoid confusion with the food environment.
I acknowledge this to be a narrative review and not a systematic review. Nevertheless, it would be good to include some information on how studies/articles/reviews included and described were selected and/or to list eg search terms that were used.
Minor note: be consistent. eg. when referring to the Eat Lancet Commission. When referring to the diet proposed by this Commission please use Planetary Health Diet (or Eat-Lancet Reference Diet)
Comments on the Quality of English LanguageThe most important issue of the manuscript is that the English text of the manuscript should be drastically improved to make it a meaningful review. In many sentences the word order is not correct or the sentence is not clear or does not make sense. I did not list all errors and issues as it would be too much.
Unfortunately, the bad English and unclear sentences distract the reader from the actual content. I recommended to have the manuscript checked and revised in terms of English language and grammar before resubmission.
Round 2
Reviewer 1 Report
Comments and Suggestions for Authors
The revised version of the manuscript has much improved in purpose and clarity. The structure is now better to follow.
There are still some minor points for consideration:
Title: It should state 'plant-based diets' and not 'plants-based diets'. Abbreviation should be used consistently, e.g., for plant-based diets (PBDs), e.g., page 3, line 121.
Comments on the Quality of English Language
English language editing should still be considered.
Reviewer 2 Report
Comments and Suggestions for Authors
The paper has significantly improved. Although it was probably the other reviewer asking for it, I also do appreciate the use of more sub-headings.
Still I have some minor comments:
Line 81: As a result --> result of what?
Line 200: please add ref for that statement
Line 230: So what? What does that result mean/imply?
Line 252: The part of empty calories come a bit out of the bleu. Please link to previous paragraph or introduce as additional issue.
Line 327 what is 'their' referring to?
Line 350 The part on processing coms a bit out of the blue. What do authors want to say here?
Line 405/406: In addition, the development of reliable sustainable diet indices is important for the valid assessment of sustainability ---> please elaborate a bit more. What should these indices cover in terms of metrics/domains?
Comments on the Quality of English Language
Also the English has very much improved but I still encountered some minor issues:
1) If you use the word nutrient or vegetable followed by intake/adequacy or so, you should not use the plural form (so it should be nutrient intake/adequacy). Examples: line 180, 181, 193, 195, 225, 272, 288
2) please don't use unnecessary words, especially if it is not entirely clear where they are referring to.
eg line 161, 175 similarly (to what?)
eg line 296 consequently
3) Other small grammar/text errors
line 33: the prevalence of obesity and overweight is highly prevalent --> Obesity and overweight are highly prevalent OR the prevalence of obesity and overweight is high.
line 49, through mediation of foods systems, one of the largest -> systems is plural so you can't use the word 'one' then
line 106 delete the word constrain
line 134 the effectiveness..... are ..., but mostly are .. --> sentence to be rephrased.
line 156 moreover --> should rather be in addition?
Lines 168 to 179 still reads strange. Consider restructuring.
line 217 & 219 the number of studies ... were --> should be was
line 228 156 countries dietary change scenarios --> 156 scenarios from different countries OR scenarios from 156 countries?
Line 400 from --> for OR in
